# How Could Consumers' Online Review Help Improve Product Design Strategy?

Wei Miao [1,2], Kai-Chieh Lin [3], Chih-Fu Wu [3], Jie Sun [4], Weibo Sun [2], Wei Wei [2,*] and Chao Gu [5,*]

1. The Graduate Institute of Design Science, Tatung University, Taipei 104, Taiwan; weimiao@cslg.edu.cn
2. School of Textile Garment and Design, Changshu Institute of Technology, Changshu 215500, China; sunweibo@cslg.edu.cn
3. Department of Industrial Design, Tatung University, Taipei 104, Taiwan; kclin@gm.ttu.edu.tw (K.-C.L.); wcf@gm.ttu.edu.tw (C.-F.W.)
4. College of Arts and Design, Zhejiang A&F University, Hangzhou 311300, China; sunjie@zafu.edu.cn
5. Academy of Arts & Design, Tsinghua University, Beijing 100084, China
* Correspondence: doublewei@cslg.edu.cn (W.W.); cguamoy@my.honam.ac.kr (C.G.)

**Abstract:** This study aims to explore the utilization of user-generated content for product improvement and decision-making processes. In the era of big data, the channels through which enterprises obtain user feedback information are transitioning from traditional methods to online platforms. The original data for this study were obtained from customer reviews of cordless hairdryers on JD.com. The specific process is as follows: First, we used the Python Requests package to crawl 20,157 initial comments. Subsequently, the initial data were cleaned, resulting in 1405 valid comments. Next, the cleaned and valid comments were segmented into Chinese words using the HanLP package. Finally, the Latent Dirichlet Allocation (LDA) method was applied for topic modeling. The visualization of the topic clustering was generated using pyLDAvis, and three optimal topics were identified. These topics were named "User Experience", "Product Evaluation", and "Product Features", respectively. Through data analysis and expert consultation, this study developed product design improvement strategies based on online reviews and verified the validity of the developed cordless hairdryer design index system through a questionnaire survey, providing practical references and innovative theoretical foundations for future product design assessments.

**Keywords:** online review; LDA; cordless hairdryers; product design improvement strategies



## 1. Introduction

Product improvement design refers to the process of making enhancements to a new generation of products based on research, analysis, and user feedback regarding their experiences and shortcomings with the existing product [1]. Traditional research methods such as telephone interviews, trial experiences, and questionnaire surveys have limitations in terms of low efficiency, small data volume, and high cost. These limitations no longer meet the needs of enterprises to obtain timely information [2,3]. In contrast, online reviews have advantages such as large data volume, easy accessibility, and wide-ranging sources. As a result, enterprises have started to focus on how to select useful online reviews and how to learn from them in order to assist in product design improvement [4]. Specifically, online reviews provide a wealth of real user experiences, and the development of natural language processing (NLP) technology enables enterprises to easily extract the core content and key issues from online reviews. In addition, the vast amount of data in online reviews has expanded their coverage, providing more comprehensive insights for product improvement. This extensive coverage allows for a more comprehensive reference for enhancing products [5]. With its abundant information and authenticity, text mining of online reviews can help businesses gain a deep understanding of how target users evaluate their products, enabling them to make targeted improvements. However,

online reviews also have certain limitations, such as insufficient representativeness of reviewers and a lack of systematic opinions. Nevertheless, businesses can still use online reviews as a complement and precursor to traditional survey methods to compensate for their shortcomings and provide more comprehensive and reliable information support for product design improvement.

Amid the rapid development of e-commerce, online review mining has emerged as a popular field [6–9]. User reviews and product purchase suggestions are also serving as crucial resources for product research and commercial realization [10–12]. This reinforces the feasibility of leveraging online reviews to acquire user experience information for product improvement [13]. While previous studies have focused on analyzing topic evolution to extract valuable consumer comments, further research is needed to explore how online reviews can support decision-making in product design improvement [14–17]. Specifically, how product designers analyze and extract insights from online reviews to identify the advantages and disadvantages of products and translate the mining results into decisions and actions for product design improvement are issues that need to be further explored.

The purpose of this study is to explore how product designers should use online reviews for data mining and decision-making when developing and optimizing products, and develop a product design index system. The remaining sections of this paper are organized as follows: Section 2 provides a literature review that traces the development and application of Latent Dirichlet Allocation (LDA) and discusses the significance of online consumer reviews in improving product design; Section 3 provides a detailed description of the research methodology and process employed in this study; Section 4 presents the topic modeling results of user review based on cordless hairdryer and develops a design index system. Finally, Section 5 emphasizes the research perspectives of this paper and provides practical recommendations for utilizing LDA in product design improvement, as well as suggestions for future research directions.

The contribution of this study lies in providing a comprehensive practical reference paradigm for utilizing LDA in product design improvement. It helps determine the priority and objectives of product design improvement based on the results of mining online product reviews. Furthermore, it provides references for translating them into actionable design specifications or design solutions. This is the significance of this research.

## 2. Literature Review

### 2.1. Development and Academic Application of LDA

Regarding the analysis methods of online product reviews, there have been numerous research achievements both domestically and internationally, which can be broadly categorized into three types:

The first type is topic evolution analysis based on word frequency statistics. This approach explores keywords representing hot topics by analyzing the characteristics of word frequency distribution in documents [18]. The initial method proposed by Kleinberg identified keywords to reveal trends in topic evolution. Some researchers have also applied bibliometric theories to track topics and influencing factors in research fields by analyzing high-frequency keywords and specialized terms [19–21]. Furthermore, some researchers utilize web crawlers to obtain word frequency data from online forums and analyze the most frequent words to capture the main issues discussed in those forums [22–24]. This method can quickly reveal the evolution of topics in a specific discipline. However, relying solely on word frequency statistics as the criterion for determining topics may not be comprehensive enough.

The second type is topic evolution analysis based on online communities, which analyzes topic evolution by identifying the community structure within a network. Girvan and Newman [25] proposed a method for identifying network communities, which has been validated and optimized in the academic community [26–29]. This method reveals topic evolution by identifying keywords within the network, with a focus on identifying highly reliable keywords. However, this approach treats all identified keywords equally

and lacks the ability to differentiate the importance of keywords. Consequently, it may not accurately represent the significance of topics, and the division of topic content may also be less reasonable.

The third type is topic evolution analysis based on topic modeling. The widely used topic model is Latent Dirichlet Allocation (LDA), proposed by Blei in 2003 [30]. LDA is an unsupervised machine learning method that can uncover latent topics from textual data. Since its introduction, LDA has garnered widespread attention and has been adopted by many researchers [31,32]. Through the use of LDA modeling, text data can be analyzed to identify topics and analyze their evolution over different time periods, revealing changes and trends. This method provides a more accurate understanding of the content and evolutionary processes of topics within textual data. Its advantages include the following:

1.  Unsupervised learning: Latent Dirichlet Allocation (LDA) is an unsupervised learning algorithm, which means that it does not require labeled data for training. Instead, it can automatically discover topics and generate topic models from unlabeled text corpora. The goal of LDA is to assign documents to latent topics by analyzing the distribution patterns of words in the text;

2.  Topic Identification and Distribution: The LDA model is highly useful in the realm of topic identification and distribution. It can be applied to large-scale collections of text to help identify hidden topics within documents and provide information on the distribution of each topic within the document. Through LDA, it is possible to obtain the relevance between each document and topic, as well as the association between each topic and word. This information is crucial for text mining and information retrieval as it aids in understanding the structure and content of a text dataset;

3.  Scalability: The LDA model exhibits excellent scalability and can handle large-scale text datasets. Traditional LDA algorithms can be accelerated through parallel computing, and they can also be trained in distributed computing environments, leveraging the computational resources of multiple machines for efficient model training. This scalability provides LDA with a significant advantage in handling large volumes of text data and enables it to tackle real-world, large-scale text mining tasks;

4.  Probabilistic Topic Modeling: LDA is a probabilistic topic modeling algorithm. It assumes that each document is composed of multiple topics, and each topic is composed of multiple words. LDA introduces probability distributions to describe the relationships between topics and words. By modeling the text data, LDA can calculate the probability distributions of each topic and each word, thereby determining the relative importance of topics. This probabilistic modeling approach enables LDA to provide a deeper understanding of the topic structure, helping us uncover the latent semantic associations within the text data.

The LDA model has undergone numerous optimizations and extensions. Blei and Lafferty [33] proposed the dynamic topic model (DTM), which incorporates document timestamps to track the evolution of topics. Chuang et al. [34] introduced a topic modeling tool based on novel visualization techniques to assess text topics. Wang et al. [35] introduced a framework that uses the LDA model to infer different topics involved in each Trump tweet and assigns a weight to each topic. These weights represent the importance or frequency of each topic in Trump's followers' tweets. By weighting and blending the weights of different topics, a comprehensive measure of topic preference is obtained to represent the degree of preference for different topics among Trump's followers on Twitter. Huang et al. [36] applied three algorithms, namely term frequency-inverse document frequency (TF-IDF), Simpson's diversity index (SDI), and Latent Dirichlet Allocation (LDA), to explore the design trends in Taiwanese design journals since the 1960s, of which the results reveal the current and future trends in the academic community, providing a reference for studying design histories in other regions of the world. Hao et al. [37] proposed a user demand acquisition method based on patent data mining. This method constructs a patent data knowledge base and applies the Latent Dirichlet Allocation topic model and K-means

algorithm to cluster patent text data, enabling the mining of key functional requirements for product development.

In summary, many scholars currently conduct topic evolution analyses based on specific domains to explore cutting-edge and trending topics. However, most of these studies focus on proposing thematic content in certain disciplines, while research on the process of product design based on user feedback and corresponding product improvements is still limited. Therefore, this study starts with the LDA model and divides the topic content into corresponding dimensions based on user needs and improvement directions. Simultaneously, it provides a more comprehensive and objective design explanation for the evolutionary path and visualization. This aims to promote theoretical innovation and application expansion in the field of data mining for product design improvements, particularly in the context of user experience feedback and iterative design processes.

### 2.2. Judgement of Topic Numbers

Determining the optimal number of topics in LDA is a challenging task. Perplexity value, coherence score, and pyLDAvis are often-used tools to assist in this process.

Perplexity is a measurement of how well a probability model predicts a sample [38]. A lower perplexity score for a certain number of topics means that the model is better at predicting the sample, suggesting that the chosen topic number is more optimal [39].The calculation of perplexity is illustrated in Equation (1).

$$Perplexity(D_{test}) = \exp\left\{ -\frac{\sum_{d=1}^{M} \log p(w_d)}{\sum_{d=1}^{M} N_d} \right\} \tag{1}$$

Note: $M$ represents the number of texts in the test corpus, $N_d$ represents the length of the d-th text (i.e., number of words), and $p(W_d)$ represents the probability of the text [30].

Previous research highlights the importance of perplexity in determining the optimal number of topics in LDA models, and the results been tested and found to be effective in various contexts. Zhao et al. [40] introduce a novel approach that uses the rate of change of perplexity as a function of the number of topics, while Gan and Qi [41] include perplexity as part of a comprehensive index for determining the optimal number of topics. However, perplexity has its limitations: it is based on the probabilistic model of LDA itself, and while it might be good at capturing the mathematical accuracy of a model, it does not necessarily align with human interpretation of topics. This is due to the fact that perplexity measures the likelihood of observing the test data of the given model, but humans judge topic models based on the interpretability of the topics, which is not necessarily correlated with the likelihood of the data [38,42]. This is where coherence comes in.

Topic coherence provides a way to evaluate the interpretability of topics by measuring the degree of semantic similarity between high scoring words in the topic [43]. The calculation of perplexity is illustrated in Equations (2)–(4).

$$\vec{v}\left(W'\right) = \left\{ \sum_{w_i \in W'} \mathrm{NPMI}\left(w_i, w_j\right)^{\gamma} \right\}_{j=1,\ldots,|W|} \tag{2}$$

$$\mathrm{NPMI}\left(w_i, w_j\right)^{\gamma} = \left( \frac{\log \frac{P\left(w_i, w_j\right) + \epsilon}{P(w_i) \cdot P(w_j)}}{-\log\left(P\left(w_i, w_j\right) + \epsilon\right)} \right)^{\gamma} \tag{3}$$

$$\phi S_i\left(\vec{u}, \vec{w}\right) = \frac{\sum_{i=1}^{|W|} u_i \cdot w_i}{\left\|\vec{u}\right\|_2 \cdot \left\|\vec{w}\right\|_2} \tag{4}$$

Note: $W$: the set of a topic's top-N most probable words $W = \{W1\ldots WN\}$. Si: a segmented pair of each word $W' \in W$ paired with all other words $W^* \in W$. S: the set of all pairs

defined as $S = \{(W', W^*) \mid W' = \{w_i\}; w_i \in W; W^* = W\}$. $P(w_i)$: probabilities of single words. $P(w_i, w_j)$: the joint probabilities of two words. $\epsilon$: coefficient accounting for the logarithm of zero. $\gamma$: coefficient to place more weight on higher NPMI values [44].

While a high coherence score suggests a more interpretable model, a low coherence score may indicate that the topics are not well-formed [45]. Therefore, coherence scores can offer a more human-centered approach to determining the optimal number of topics, as they aim to align more closely with how humans interpret topics [46]. Coherence scores such as UMass and UCI are often used in conjunction with perplexity to determine the optimal number of topics [30,47].

Lastly, pyLDAvis is a Python library for interactive topic model visualization, which can be an invaluable tool for interpreting the output of an LDA model [48]. This tool can provide a more intuitive understanding of the topics that the model has learned. PyLDAvis generates a two-dimensional map of the topics, where the distance between topics indicates their similarity. It also provides a list of the most relevant terms for each topic, which can be sorted by their frequency or overall relevance. This allows users to easily explore the main terms associated with each topic and understand their context in the model. The use of pyLDAvis adds a visual dimension to the model evaluation process, making it easier to understand the relationships between different topics and the terms that constitute them. It also helps in identifying potential issues in the model, such as overlapping topics. For instance, if two topics are very close in the pyLDAvis space and have many shared terms, it might be an indication that the number of topics is too high. Conversely, if topics are well-separated and have distinct sets of terms, this could suggest that the model has identified meaningful structure in the data.

To conclude, perplexity value, coherence score, and pyLDAvis each offer unique advantages in assisting the determination of the optimal number of topics for LDA topic modeling. Perplexity value provides a statistical measure of the model's predictive ability, coherence score offers a more human-centered evaluation of topic interpretability, and pyLDAvis provides a visual, intuitive understanding of the topics and their relationships. By using these tools in conjunction, this study tries to better ensure the accuracy and interpretability of LDA models, thereby improving the quality of topic modeling results.

### 2.3. Online Consumer Reviews

In the current e-commerce environment, online consumer reviews (OCRs) have become an important source of information for businesses to acquire insights about product performance and consumer needs [49]. These reviews not only help businesses understand the strengths and weaknesses of existing products but also provide valuable references for the development of new products.

Sparks et al. [50] point out that online reviews are a persuasive means of influencing consumer behavior. Consumers perceive specific information from other consumers as the most useful and trustworthy, which may be one reason why they are willing to share their shopping experiences. Additionally, some consumers are aware that these reviews can influence the purchasing decisions of others. Positive reviews, descriptive ratings, image reviews, additional comments, and cumulative reviews all impact consumer buying behavior. These findings collectively indicate that consumers are willing to share their online shopping experiences because they recognize the value of these reviews to other consumers and their ability to influence others' purchasing decisions (Mo et al.) [51].

Cheng et al. [52] utilized user-generated online reviews to objectively evaluate the user experience quality of existing products and provide detailed recommendations for product improvement. Zhao et al. [53] proposed a consumer satisfaction modeling method based on an improved Kano model using online review data, which assists product improvement by considering risk attitude and expectations. Therefore, mining and analyzing online reviews can help businesses understand consumers' risk attitudes and expectations toward product attributes, thereby better meeting consumer needs in new product development.

## 3. Methodology

### 3.1. Data Source and Collection

The study focuses on cordless hairdryers, which have become a popular category in the personal care product market in recent years. Users have various expectations and feedback regarding the functionality, performance, and user experience of cordless hairdryers. As an emerging product category, there is a demand for design improvements. Therefore, collecting online reviews of cordless hairdryer products from online shopping websites and analyzing them using the LDA model can extract valuable information from a large amount of user feedback. This information can serve as a scientific basis for product improvement and strategy optimization. The methodology employed in this study combines user requirements with data analysis, enabling businesses to better understand the market and users, thereby enhancing product competitiveness and user satisfaction. Additionally, JD.com serves as one of the largest 3C e-commerce platforms in China, attracting a vast number of consumers and products. Therefore, the user reviews and feedback data on cordless hairdryers available on this platform can provide rich information to analyze consumers' needs and preferences for different brands and models.

The data collection was conducted on 16 February 2023. The search keyword used was "cordless hairdryer", and the nine highest-reviewed products were selected from JD.com. User comments on the JD.com website are publicly visible to all users, providing reference for both sellers and other users for exchanging product details. The comment section also facilitates interaction among buyers, allowing them to exchange information about the products. Therefore, the data used in this study are obtained from a legitimate source.

### 3.2. Data Processing

The data processing consists of four steps: data collection, data cleaning, topic modeling, and formulating product design guidelines, as illustrated in Figure 1.

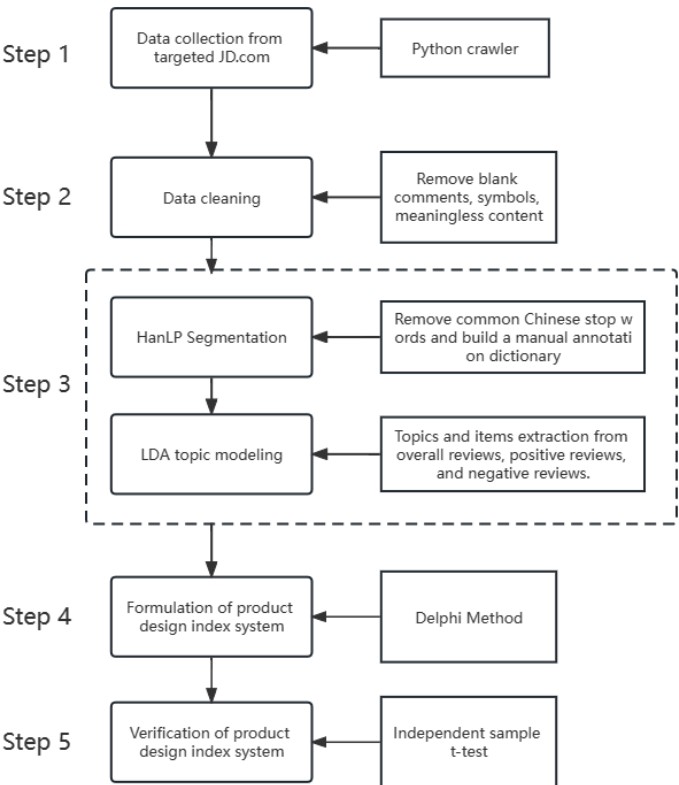

**Figure 1.** Methods and research procedure.

First, this study used the Python Requests package to crawl the reviews of the top nine cordless hairdryer products in JD.com, with a total of 20,157 initial review data. The raw data included user ID, user nickname, product review content, comment time, and other information. To protect user privacy, this study only used the product review content for subsequent research analysis.

After inspecting the original review content, this study found a large number of default reviews, blank reviews, and insufficient expressions such as "good", "excellent", and "bad" in the raw data. It was difficult to obtain useful information from these reviews. Therefore, this study removed these invalid reviews and obtained 1405 valid comments.

The third stage mainly consisted of two parts: Chinese text segmentation and formal topic analysis. To conduct topic modeling, the complete text messages needed to be segmented into keywords through word segmentation. This study used the HanLP library for text segmentation. HanLP is a popular third-party Chinese word segmentation library. It uses its own basic Chinese word library to determine the probability of association between Chinese characters. It forms word combinations based on high association probabilities, resulting in word segmentation results. Furthermore, since the main data in this study came from user reviews of cordless hairdryers, three scholars specializing in product design were invited to manually annotate the specialized vocabulary and corresponding parts of speech in the text messages. Based on these vocabulary and parts of speech, user-defined dictionaries were further defined for the HanLP library to enable more accurate text segmentation of the messages. To avoid the impact of meaningless words or characters on subsequent topic modeling analysis, this study removed common stop words from the text messages before formal topic modeling. "Stop words" refers to certain words or terms that are automatically filtered out before or after processing natural language data in order to save storage space and improve search efficiency in information retrieval. These stop words are manually inputted and do not come from automatic generation. Stop word lists are the commonly used tool to remove stop words from texts. However, there is no definitive stop word list that can be applied to all cases. In this study, we integrated several commonly used stop word libraries in Chinese natural language processing, including the HIT (Harbin Institute of Technology) stop word list, Baidu stop word list, Sichuan University Machine Intelligence Laboratory stop word list, and common Chinese stop word list.

Next, Latent Dirichlet Allocation (LDA) was used for topic analysis. LDA provides a probability distribution of topics for each document in a document collection. By analyzing the topic distributions of some documents and extracting their topic distributions, topic clustering can be performed based on these topic distributions. In this study, the importance of each topic term was determined using Term Frequency-Inverse Document Frequency (TF-IDF) weight data. Through an accurate understanding of the true intention of user reviews, errors in judgment based on individual words were avoided. TF-IDF is a statistical method used to evaluate the importance of a term in a document collection or corpus [54–56]. The importance of a term increases proportionally with its frequency in the file but is inversely proportional to its frequency in the corpus. Various forms of TF-IDF weighting are commonly used in search engines as measures or ratings of the relevance between files and user queries.

In addition, this study develops a design index system for wireless hair dryers after collecting online evaluation data of the products, using the LDA model for topic modeling of the evaluation texts, and organizing user requirements by converting product features into user demands. The Delphi method is a structured decision support technique aimed at obtaining relatively objective information, opinions, and insights through independent and iterative subjective judgments from multiple experts during the information-gathering process [57]. Hwang et al. [58] achieved good performance by introducing expert analysis in their study on topic modeling of online user emotional eating. Aranda et al. [59] proposed a mixed method that combines critical discourse analysis with structural topic modeling, transforming the challenges reflected by the topic vocabulary into valuable opportunities. Therefore, this study develops the design index system for cordless hairdryers using the

Delphi method. By comparing the product feature words and comparative text review content and exploration of user needs, a design index system for cordless hairdryers based on user needs was developed, and product improvement suggestions for cordless hairdryer design were proposed.

Finally, this study verifies the developed cordless hairdryer design index system through a questionnaire survey. We developed a questionnaire to assess whether there were significant differences in consumer satisfaction, purchase intention, usage habits, and intention to continue using of the selected cordless hairdryer samples through independent sample *t*-test to verify the validity of the cordless hairdryer design index system.

## 4. Results and Discussion

### 4.1. Topic Modeling and Visualization of User Comments

This study uses LDA topic modeling to mine user needs lurking in the comments of cordless hairdryer products. Through a Python crawler, a total of 20,157 initial comments were collected in the initial stage. After removing reviews with preset content, blank reviews and those with only one word, only 1405 reviews were deemed valid and suitable for subsequent analysis. This study presents a descriptive statistical analysis of the review samples. The longest review in the sample comprises 329 words, while the shortest review consists of only two words. The average word length of the reviews is calculated to be 50.103 words. The reviews with a length of 55 words exhibit the highest frequency, occurring a total of 39 times. Additionally, the standard deviation of the total valid review samples is calculated to be 32.746 words.

According to the conclusions drawn from the literature discussion above, this study explores the optimal number of topics by manually adjusting the number of topics and evaluating perplexity value and coherence score. The pyLDAvis package is employed to assist in determining the optimal number of topics. In this study, perplexity values (Figure 2), coherence scores (Figure 3), and pyLDAvis clustering results of LDA models ranging from 1 to 10 topics were calculated. The results indicate that the perplexity value ranges between 60 and 80 and gradually increases with the number of topics, suggesting that an increasing number of topics leads to poorer topic modeling results. On the other hand, the coherence score fluctuates between 0.3 and 0.4, with higher coherence scores indicating better topic modeling outcomes. By adhering to the principle of minimizing perplexity while maximizing coherence and considering the visualization results from pyLDAvis (Figure 4), the optimal number of topics is determined to be three.

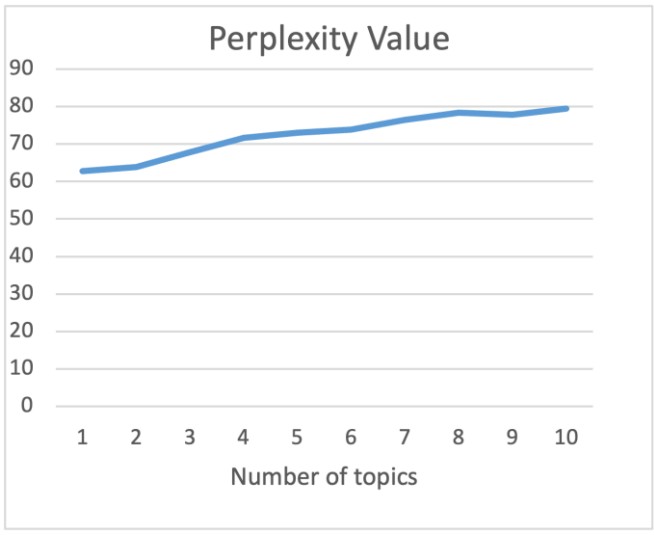

**Figure 2.** Results of perplexity value.

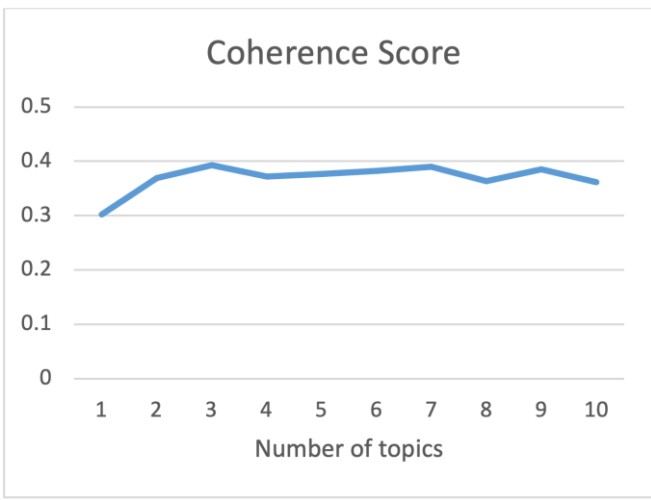

**Figure 3.** Results of coherence score.

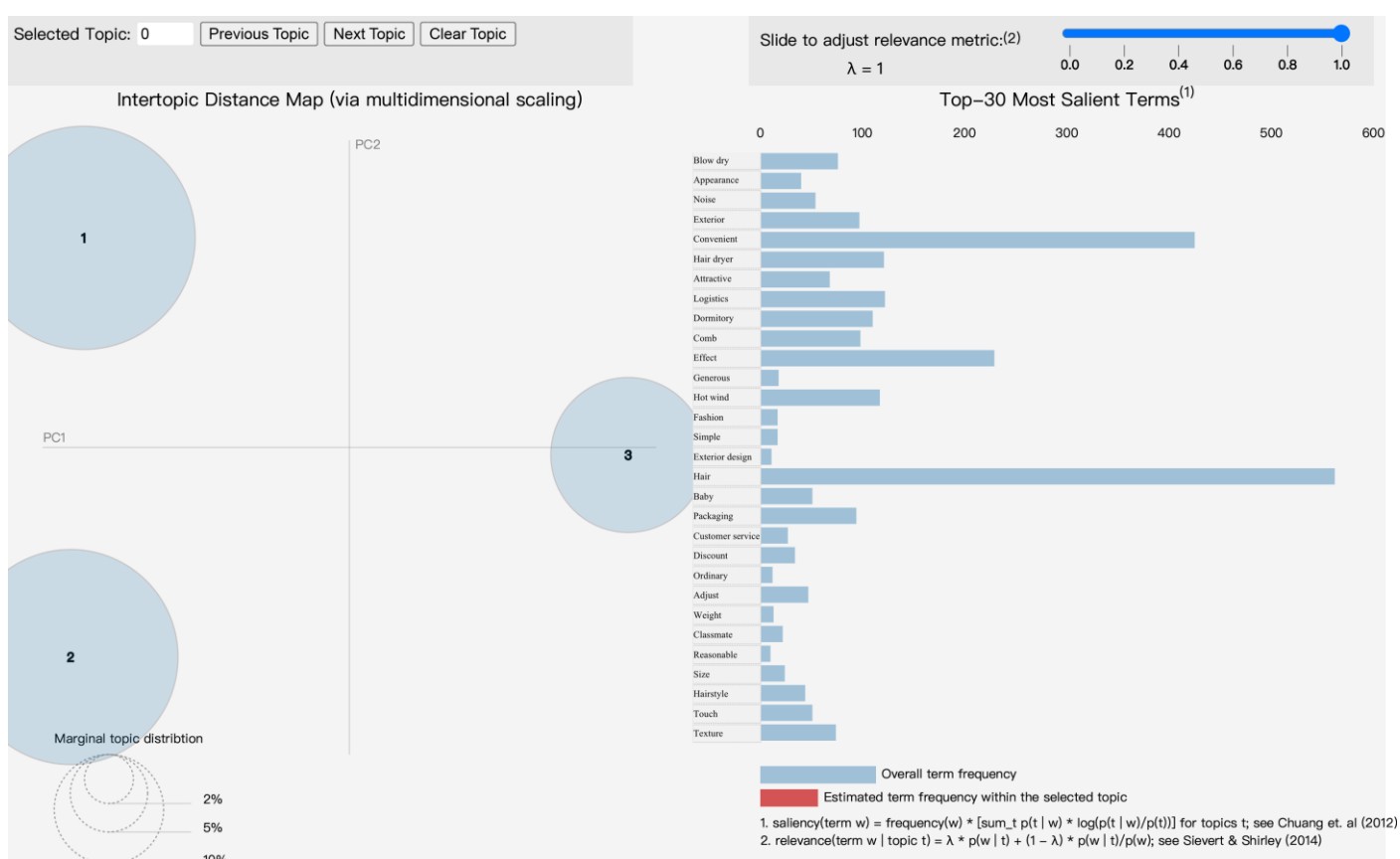

**Figure 4.** PyLDAvis results of 3 topic modeling [34,48].

### *4.2. Topic Model and Labeling*

This study uses LDA to conduct topic modeling of users' online comments of cordless hairdryer. Table 1 shows the topic modeling results of the top ten high-frequency words within each topic when there are three topics.

**Table 1.** Top 10 high-frequency words of 3 topics modeling.

| Total Documents | Proportion of Documents Supported | Top 10 High-Frequency Words |
|---|---|---|
| 1405 | 46% | Not bad, convenient, straight hair, children, express delivery, hot air, effect, easy, feeling, operation |
| | 27.5% | Effect, quality, speed, wind force, drying, positive comments, curls, essential oil, activities, price |
| | 26.4% | Dormitory, appearance, practical, noise, time, speed, shape, temperature, tactile, aesthetics |

In order to better present the key words in the comments, this study uses TF-IDF (Term Frequency-Inverse Document Frequency) to measure the importance of a word in the collected reviews. TF stands for term frequency, which refers to the frequency of a word appearing in a document. IDF stands for inverse document frequency, which represents the importance of a word in the entire document collection. The TF-IDF value indicates the significance of a word in a document. If a word appears frequently in a document (high TF), but rarely in the entire document collection (high IDF), its TF-IDF value will be high, indicating that the word is highly important in the document. The TF-IDF results are presented in Appendix B.

According to the key words and word cloud visualization of the analysis results, this study invited five product design experts to summarize the topics and explain the corresponding requirements, so as to facilitate the development of the cordless hairdryer design index system in the next step (Figure 5).

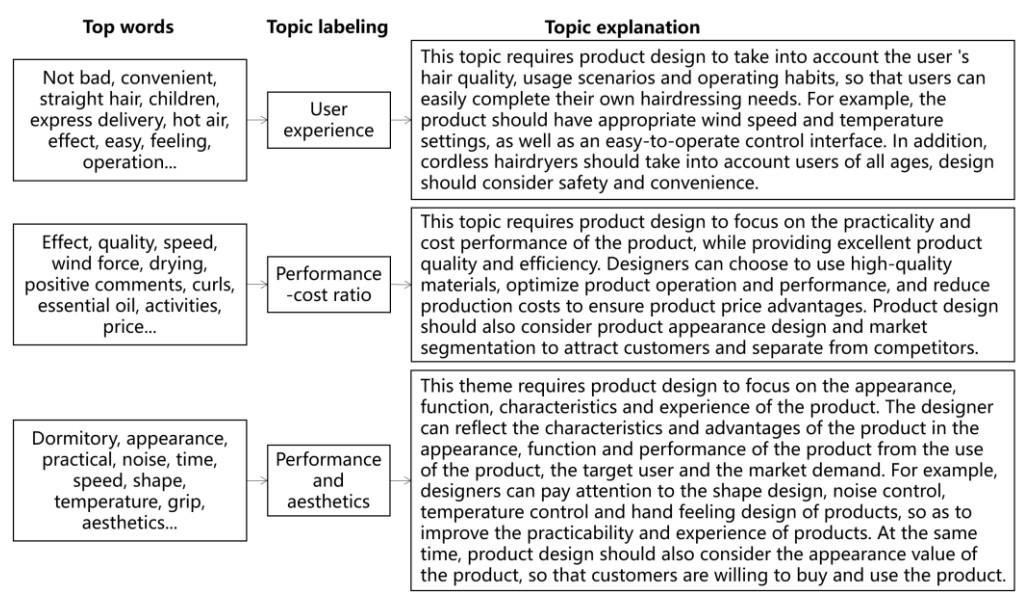

**Figure 5.** Topic clustering and description of online comments on cordless hairdryers.

*4.3. Development of Cordless Hairdryer Design Index System Based on User Needs*

Through the process of collecting online comments of cordless hairdryer products, using LDA modeling to identify the topic of commentary text, and mapping product feature words to user needs, user needs are gradually excavated, and the design index system of cordless hairdryers is developed based on user needs.

We invited five product design experts (see Table 2), aged between 38 and 55, with an average industry experience of over 15 years, to integrate and elaborate on these themes and related requirements.

**Table 2.** Information of invited experts.

| No. | Specialty | Age | Position | Length of Employment |
|-----|-----------|-----|----------|----------------------|
| 1 | Design engineering | 38 | Project Manager | 10 |
| 2 | Industrial design | 39 | Project Manager | 10 |
| 3 | Industrial design | 43 | Professor | 14 |
| 4 | Industrial design | 46 | Professor | 16 |
| 5 | Design engineering | 55 | Senior Researcher | 27 |

The specific steps of the research are as follows: We organized the results of the LDA analysis in a visual format, by creating a table that lists the keywords and phrases under each topic. Additionally, we prepared background information for each product group, including basic product details, target user groups, and market environment, to facilitate the experts' deeper understanding of the thematic context. Then we invited five product design experts to participate in a group discussion, followed by individual in-depth conversations. Before their attendance, we explained the purpose of the discussion to the experts and clarified the tasks they needed to complete. Once the discussion commenced, we presented the LDA analysis results to the experts and invited them to integrate the various topics. This process requires the experts to comprehend and interpret the topics before consolidating each topic into one or more product requirements. For example, if a topic includes keywords like "good" and "convenient", the experts may integrate it into a requirement for "product usability". After the experts integrated the requirements, they needed to provide detailed explanations for each requirement, including the specific content, possible implementation approaches, and potential challenges during the implementation process. The purpose of this step was to ensure a comprehensive understanding and consideration of each requirement. Once the experts completed the theme integration and requirement elaboration, we began developing the design index system for the cordless hairdryer (see Table 3). This system encompasses all the requirements and specify the indicators for fulfilling each requirement. For example, if a requirement is "product usability", a corresponding indicator might be "design should meet user needs based on the scenarios in which the product is used, such as travel or home use".

It can be seen that the cordless hairdryer should have adaptability in terms of user experience, such as providing different temperature and wind speed settings, considering the usage scenario, as well as easy operation and safety. In terms of product evaluation, practicality and quality are crucial considerations for the wireless hairdryer, such as speed and airflow. High-quality materials should be used to ensure product quality and performance while reducing production costs to maintain a competitive price advantage. Additionally, attention should be given to user feedback and evaluations to understand the product's performance in the market. Regarding product features, the wireless hairdryer's design should meet fashion and aesthetic standards, incorporate advanced technology to reduce noise and provide a better user experience. Adjustable temperature control should be provided to cater to different user needs. Furthermore, the tactile design of the product should be considered to ensure user comfort.

Based on the aforementioned discussion, the expert group, through the thematic word associations and contextual evaluations, provided constructive suggestions for design indexes through multiple rounds of discussions. These suggestions include optimizing the airflow control function, improving battery life, adopting lightweight design, and enhancing the tactile experience of the product. Furthermore, during the design process, close attention should be paid to market demands and user feedback to continuously optimize and improve the product design, ensuring it meets user needs and remains competitive.

**Table 3.** Cordless hairdryer design index system.

| Topic | Indexes | Explanation |
|---|---|---|
| User experience | Adaptability | The product should be adaptable to the needs of users with different hair types, such as providing different temperature and wind speed settings. |
| | Usage scenarios | To design a product that meets user needs, considering the scenarios in which users will use the product, such as travel or home use, is essential. |
| | Ease of use | Product design should consider a user-friendly control interface that facilitates ease of use for the users. |
| | Safety | Considering the needs of children using the product, the design should prioritize both safety and convenience. |
| Performance-cost ratio | Practicality | The practicality and performance of the product should receive attention, including factors such as speed and power. |
| | Product quality | Using high-quality materials is essential to ensure product quality and performance. |
| | Price advantage | To ensure price advantage, it is important to reduce product costs while considering the product's appearance and target market segment to attract customers. |
| | User reputation | Considering user reviews and feedback is crucial to understand how the product is performing in the market. |
| Performance and aesthetics | Appearance design | Emphasizing the exterior design of the product to align with fashion and aesthetic standards is important. |
| | Noise control | Adopting advanced technology to reduce noise and provide a better user experience is essential. |
| | Temperature control | Providing adjustable temperature control is essential to meet the diverse needs of different users. |
| | Tactile feel design | Considering the tactile design of the product is crucial to ensure user comfort. |

*4.4. Cordless Hairdryer Design Index System Verification*

This phase seeks to ascertain whether the chosen products based on the cordless hairdryer design index system developed in this study align with the evaluations made by consumers. The previous expert group (Table 2) was invited to select the top 20 products from the Jingdong website according to the cordless hairdryer design index system. In order to eliminate the consumer evaluation bias caused by different grades of products, this study selected a pair of products with relatively close prices and functions from 20 products. Among the products, one (labeled Sample 2 in Table 4) is a product that conforms to the cordless hairdryer design index system developed in this study, and the other is a product that is less compliant (labeled Sample 1 in Table 4). In addition, in order to avoid being overly influenced by brand, color or other factors, the study processed images and added some text descriptions to help respondents focus on the evaluation of product functionality and usability.

**Table 4.** Product Sample and Details.

| Sample | Details |
| --- | --- |
| 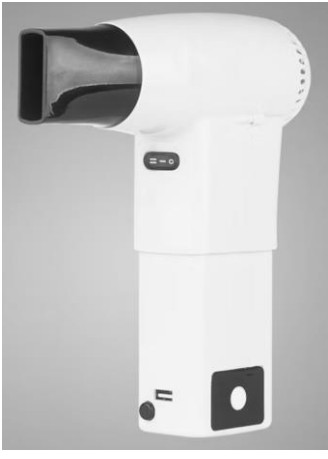<br>Sample 1 | Advantages: The product basically meets the needs of cordless hairdryer in terms of adaptability, operation convenience and practicability.<br>Disadvantages: The evaluation of product characteristics is low, especially in appearance design and hand feeling design. In addition, the security design is relatively simple, and the user reputation is general. |
| 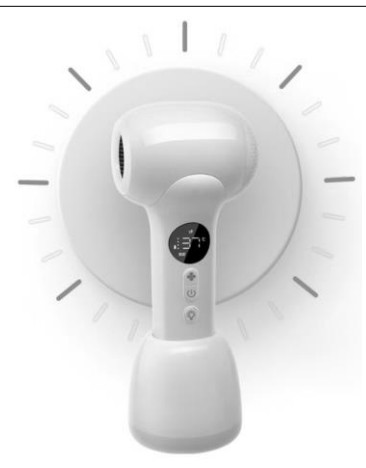<br>Sample 2 | Advantages: The product has been recognized by experts in all aspects of the evaluation subject.<br>Disadvantages: The power of the product is relatively small, which is a common technical difficulty in cordless hairdryers. |

Appendix A shows the scale items we used for verification. The questionnaire includes four parts: user satisfaction, purchase intention, usage habits, and intention to continue using. The questionnaire in this study is modified on the basis of the mature scale verified in the previous study [60–63]. The questionnaire is a 5-point Likert's scale. Before the formal test, this study invited five randomly selected respondents who met the research requirements to conduct a questionnaire pre-test. As part of the evaluation process, they were asked to assess whether they fully understood the content of each questionnaire item. After the pre-test was completed, the expression of the questionnaire was modified according to the feedback of the interviewees to improve readability. During the formal test phase, all respondents were confirmed to have used such products and fully understood the function and use of the two samples before starting to fill in the questionnaire. The questionnaire survey was conducted in July 2023, and a total of 290 questionnaires were collected. In order to ensure the validity of the respondents' feedback, we set up reverse questions and judged their concentration according to the time they answered the questions. After excluding invalid samples, a total of 244 valid questionnaires were collected in this study. The demographic characteristics of the respondents are shown in Table 5.

**Table 5.** Demographic characteristics of the respondents.

| | Category | Count | Ratio (%) |
|---|---|---|---|
| Gender | Male | 119 | 48.77 |
| | Female | 125 | 51.23 |
| Age | 20 or younger | 13 | 5.33 |
| | 21–29 | 88 | 36.07 |
| | 30–39 | 113 | 46.31 |
| | 40–49 | 20 | 8.20 |
| | 50–59 | 9 | 3.69 |
| | 60 or older | 1 | 0.41 |
| Education | Junior high school or below | 2 | 0.82 |
| | High school or secondary school | 9 | 3.69 |
| | Undergraduate or college | 194 | 79.51 |
| | Postgraduate or higher | 39 | 15.98 |
| Marriage Status | Unmarried | 90 | 36.89 |
| | Married | 154 | 63.11 |
| Monthly Income | 3000 or less | 46 | 18.85 |
| | 3001–5000 | 30 | 12.30 |
| | 5001–8000 | 57 | 23.36 |
| | 8001–12,000 | 67 | 27.46 |
| | 12,001 or more | 44 | 18.03 |
| Occupation | Professionals (such as teachers/doctors/lawyers, etc.) | 32 | 13.11 |
| | Service workers (catering waiter/driver/salesperson, etc.) | 3 | 1.23 |
| | Freelancers (such as writers/artists/photographers/tour guides, etc.) | 2 | 0.82 |
| | Workers (such as factory workers/construction workers/urban sanitation workers, etc.) | 1 | 0.41 |
| | Staff | 133 | 54.51 |
| | Public institutions/civil servants/government workers | 16 | 6.56 |
| | Student | 51 | 20.90 |
| | Other | 6 | 2.46 |
| Area | North China: Beijing, Tianjin, Hebei, Shanxi, Inner Mongolia | 38 | 15.57 |
| | Northeast China: Liaoning, Jilin, Heilongjiang | 17 | 6.97 |
| | East China: Shanghai, Jiangsu, Zhejiang, Anhui, Fujian, Jiangxi, Shandong | 105 | 43.03 |
| | Central China: Henan, Hubei, Hunan | 23 | 9.43 |
| | South China: Guangdong, Guangxi, Hainan | 43 | 17.62 |
| | Southwest China: Chongqing, Sichuan, Guizhou, Yunnan, Tibet | 14 | 5.74 |
| | Northwest China: Shaanxi, Gansu, Qinghai, Ningxia, Xinjiang | 1 | 0.41 |
| | Hong Kong, Macao, Taiwan | 3 | 1.23 |

In order to verify the validity of the cordless hairdryer design index system, we used SPSS 26 for data analysis to assess whether there were significant differences in consumer satisfaction, purchase intention, usage habits, and intention to continue using of the selected cordless hairdryer samples. This study used Cronbach's $\alpha$ test to evaluate the reliability of the questionnaire data. The results show that the Cronbach's $\alpha$ value of the user satisfaction dimension is 0.853, the Cronbach's $\alpha$ value of the purchase intention dimension is 0.875, the Cronbach's $\alpha$ value of the usage habits dimension is 0.875, and the Cronbach's $\alpha$ value of the intention to continue using dimension is 0.863. The Cronbach's $\alpha$ value of each dimension is greater than 0.85, and the Cronbach's $\alpha$ value will not be higher than the current value after deleting any item within the scales, indicating that the data of this study are reliable and suitable for subsequent analysis [64]. In this study, Levene and Box tests were used to determine whether the data distribution met the prerequisites of multivariate analysis of variance (MANOVA). The Levene's test results showed that user satisfaction (Levene Statis-

tic = 9.165, $p < 0.05$), purchase intention (Levene Statistic = 21.092, $p < 0.05$), usage habits (Levene Statistic = 5.497, $p < 0.05$), intention to continue using (Levene Statistic = 6.793, $p < 0.05$). The Box test also showed significant results (Box's M = 90.945, F = 2.986, $p < 0.05$). Hence, the data do not meet the MANOVA prerequisites, so we use the independent sample $t$-test to conduct difference analysis.

The results of the independent sample $t$-test (Table 6) showed that the average scores between Sample 1 and Sample 2 were significantly different in the four dimensions of consumer satisfaction, purchase intention, usage habits, and intention to continue using ($p < 0.05$), and the evaluation of Sample 2 in each dimension was higher than those of Sample 1, which was in line with the evaluation results of the experts based on the cordless hairdryer design index system. We also tested whether there were differences in the four dimensions between different gender users (Table 7). All the results were not significant ($p > 0.05$), indicating that there were no significant differences in perception between men and women in these four dimensions. In addition, we compared the interaction effect between samples and gender (Table 8). The results showed that samples and gender did not produce significant interaction effect on any of the four dimensions, which showed that men and women have similar views and needs when evaluating the design indicators of cordless hairdryer products, and also showed that the cordless hairdryer design index system developed in this study had good versatility.

**Table 6.** Independent Sample $t$-Test (Category).

| Dimension | Category | Mean | Standard Deviation | t | Significance (2-Tailed) |
|---|---|---|---|---|---|
| SA | Sample 1 | 3.62842 | 0.870380 | −7.102 | 0.000 |
| | Sample 2 | 4.13525 | 0.696464 | | |
| PI | Sample 1 | 3.54372 | 1.119067 | −5.985 | 0.000 |
| | Sample 2 | 4.05874 | 0.744872 | | |
| UH | Sample 1 | 3.26913 | 1.091168 | −4.951 | 0.000 |
| | Sample 2 | 3.71175 | 0.871365 | | |
| CUI | Sample 1 | 3.38388 | 1.051354 | −4.978 | 0.000 |
| | Sample 2 | 3.81831 | 0.867675 | | |

**Table 7.** Independent Sample $t$-Test (Gender).

| Dimension | Gender | Mean | Standard Deviation | t | Significance (2-Tailed) |
|---|---|---|---|---|---|
| SA | Male | 3.87535 | 0.850173 | −0.169 | 0.866 |
| | Female | 3.88800 | 0.806537 | | |
| PI | Male | 3.80952 | 0.967422 | −0.182 | 0.856 |
| | Female | 3.79333 | 1.001316 | | |
| UH | Male | 3.50280 | 1.003856 | −0.263 | 0.792 |
| | Female | 3.47867 | 1.019549 | | |
| CUI | Male | 3.58403 | 0.976127 | −0.372 | 0.710 |
| | Female | 3.61733 | 0.999224 | | |

**Table 8.** Demographic characteristics of the respondents.

| Source | Dependent Variable | Type III Sum of Squares | Mean Square | F | Significance (2-Tailed) |
|---|---|---|---|---|---|
| Category × Gender | SA | 0.439 | 0.439 | 0.704 | 0.402 |
| | PI | 0.519 | 0.519 | 0.573 | 0.450 |
| | UH | 0.767 | 0.767 | 0.785 | 0.376 |
| | CUI | 0.938 | 0.938 | 1.008 | 0.316 |

## 5. Conclusions and Limitations

Based on the above analysis and discussions, this study proposes and validates a comprehensive method for utilizing user review content based on LDA for product design improvement. Ultimately, the cordless hairdryer design index system based on user needs has been developed and verified. The development process of a cordless hairdryer product index based on LDA text mining and the Delphi method proposed by this study is reasonable and effective. The results of this study can serve as a good reference for the improvement of the development process of cordless hairdryer.

### 5.1. Theoretical Implications

This study introduces a multi-reference judgment method utilizing perplexity value, coherence score, and pyLDAvis to determine the optimal number of topics in LDA topic modeling. This approach aids in the determination of the number of topics during the LDA topic modeling process. Chinese, compared to many other languages such as English, exhibits significant differences in grammar and structure [65]. For instance, Chinese lacks explicit word boundaries (such as spaces), requiring the development of dictionaries and the selection of appropriate stop word lists during the preliminary data processing phase. Moreover, Chinese exhibits a high degree of polysemy and ambiguity, which may affect the accuracy of the analysis [66]. Therefore, the results of LDA need to be cross-checked and interpreted using other methods such as Delphi analysis. Additionally, the research workflow of this study can serve as a reference for applying LDA in product design improvement.

### 5.2. Practical Implications

This study conducts topic modeling on customer review data from JD.com, and develops a product design improvement index system based on online reviews. The system primarily focuses on user experience, product evaluation, and product features. When developing cordless hairdryers, manufacturers not only need to consider user requirements regarding temperature, wind speed, operation, and safety in different usage scenarios but also need to balance practicality, price, and quality. Additionally, aspects such as appearance design, tactile feedback, and noise levels should also be taken into account. In addition, manufacturers should pay more attention on the functional and practical attributes, and it will not be helpful to take gender difference into consideration.

### 5.3. Limitations and Future Studies

It is worth noting that there are some limitations in using the LDA model for product design evaluation. For instance, the LDA model often only considers the company's own products and overlooks comparisons with competing products, potentially leading the company in the wrong direction for product improvement [67]. Furthermore, the LDA model may not effectively handle multi-group or multilingual text analysis and has limited capabilities for sentiment analysis within the text [68]. Therefore, when applying the LDA model for product design evaluation, it is important to consider its limitations and select appropriate models for in-depth analysis and exploration.

Finally, it is discovered that applying the LDA model enables the analysis of topic evolution and trends in review data. This helps identify directions for product improvement and optimization opportunities. By observing how topics change over time, one can understand the evolving perspectives and changing trends in consumer opinions and needs regarding the product.

In the future, it is suggested to combine other algorithms and methods for a more comprehensive product design evaluation. For example, sentiment analysis can be used to identify the emotional tendencies in reviews, providing better understanding of consumer emotional feedback. Additionally, machine learning classification algorithms can be employed to automatically categorize reviews, enabling a more detailed analysis of different types of feedback and suggestions.

**Author Contributions:** Conceptualization, W.M. and W.W.; methodology, C.-F.W.; software, C.G.; validation, W.M., C.G and W.W.; formal analysis, W.S.; investigation, K.L; resources, J.S.; data curation, C.G.; writing—original draft preparation, W.M.; writing—review and editing, W.W.; visualization, C.G.; project administration, K.-C.L. All authors have read and agreed to the published version of the manuscript.

**Funding:** This research received no external funding.

**Data Availability Statement:** All the original research data can be obtained by emailing correspondence author.

**Conflicts of Interest:** The authors declare no conflict of interest.

## Appendix A

The items used for the verify are shown in Table A1.

**Table A1.** The scale of measurement.

| Dimensions | Items | Source |
|---|---|---|
| User satisfaction | I'm very satisfied with this cordless hairdryer. | [60] |
| | The quality of this cordless hairdryer is very good. | |
| | This cordless hairdryer meets my expectations. | |
| Purchase intention | It is likely for me to purchase this cordless hairdryer. | [61] |
| | I am capable of purchasing this cordless hairdryer. | |
| | It is possible for me to purchase this cordless hairdryer. | |
| Usage habits | I would like to stop using my current cordless hairdryer and use this one. | [62] |
| | My family would like to stop using my current cordless hairdryer and use this one. | |
| | My friends would like to stop using my current cordless hairdryer and use this one. | |
| Intention to continue using | I would like to continue my usage of this cordless hairdryer. | [63] |
| | I would like to continue my usage of this cordless hairdryer rather than stop. | |
| | My intentions are to continue using this cordless hairdryer rather than use any alternative means. | |

## Appendix B

**Table A2.** Results of TF-IDF Calculation.

| Word | Occurrence | Reviews | TF | TF-IDF |
|---|---|---|---|---|
| Hair | 598 | 534 | 0.036299624 | 0.015262768 |
| Very | 301 | 278 | 0.018271215 | 0.012865541 |
| Straight hair | 308 | 296 | 0.018696127 | 0.012657098 |
| Not bad | 317 | 330 | 0.019242443 | 0.012103162 |
| Convenient | 365 | 415 | 0.022156125 | 0.011744016 |
| Fast | 253 | 283 | 0.015357533 | 0.010692069 |
| Effect | 221 | 260 | 0.013415078 | 0.009834679 |
| Use | 180 | 187 | 0.010926308 | 0.009574172 |

**Table A2.** *Cont.*

| Word | Occurrence | Reviews | TF | TF-IDF |
|---|---|---|---|---|
| Very good | 187 | 206 | 0.01135122 | 0.00946438 |
| Special | 161 | 161 | 0.009772976 | 0.009188862 |
| TRUE | 167 | 183 | 0.010137186 | 0.008974109 |
| Useful | 153 | 164 | 0.009287362 | 0.008668753 |
| Heat | 153 | 177 | 0.009287362 | 0.008365343 |
| Appearance | 137 | 155 | 0.008316135 | 0.007965335 |
| Seems | 137 | 159 | 0.008316135 | 0.007876882 |
| Quality | 133 | 156 | 0.008073328 | 0.007703913 |
| Comb | 112 | 115 | 0.006798592 | 0.007385717 |
| Very useful | 125 | 151 | 0.007587714 | 0.007350373 |
| Operation | 118 | 145 | 0.007162802 | 0.007073438 |
| Recommendation | 113 | 132 | 0.006859293 | 0.007049743 |
| Receive | 110 | 129 | 0.006677188 | 0.006919443 |
| Children | 102 | 107 | 0.006191575 | 0.006911798 |
| Like | 108 | 124 | 0.006555785 | 0.006908687 |
| Speed | 110 | 131 | 0.006677188 | 0.006890873 |
| Jingdong | 106 | 123 | 0.006434381 | 0.006809706 |
| Will not | 102 | 115 | 0.006191575 | 0.006726278 |
| Express delivery | 104 | 131 | 0.006312978 | 0.006515007 |
| Comparison | 98 | 114 | 0.005948768 | 0.006491369 |
| Easy | 99 | 122 | 0.006009469 | 0.006387337 |
| Feeling | 98 | 120 | 0.005948768 | 0.006350158 |
| Temperature | 94 | 109 | 0.005705961 | 0.006340369 |
| Packaging | 91 | 106 | 0.005523856 | 0.006195138 |
| Did not | 89 | 102 | 0.005402452 | 0.006145356 |
| Time | 88 | 101 | 0.005341751 | 0.006105489 |
| Curls | 88 | 102 | 0.005341751 | 0.006076307 |
| Natural | 86 | 100 | 0.005220347 | 0.005995609 |
| Exactly | 75 | 84 | 0.004552628 | 0.005559025 |
| Hair curler | 74 | 83 | 0.004491927 | 0.005514689 |
| Things | 73 | 82 | 0.004431225 | 0.005470003 |
| Satisfaction | 72 | 84 | 0.004370523 | 0.005336664 |
| No need | 71 | 88 | 0.004309822 | 0.005179342 |
| Purchase | 67 | 75 | 0.004067015 | 0.005164093 |
| Product | 66 | 74 | 0.004006313 | 0.00511676 |
| Price | 69 | 84 | 0.004188418 | 0.005114303 |
| Inward curls | 68 | 82 | 0.004127716 | 0.005095345 |
| Hot wind | 67 | 81 | 0.004067015 | 0.00504823 |
| Hair style | 67 | 83 | 0.004067015 | 0.004993029 |
| Frizzy | 68 | 87 | 0.004127716 | 0.004986667 |
| White | 66 | 86 | 0.004006313 | 0.004865777 |

**Table A2.** *Cont.*

| Word | Occurrence | Reviews | TF | TF-IDF |
|---|---|---|---|---|
| Hot to touch | 62 | 74 | 0.003763506 | 0.004806654 |
| After washing | 63 | 78 | 0.003824208 | 0.004800423 |
| Straighten | 63 | 78 | 0.003824208 | 0.004800423 |
| Essential oil | 62 | 77 | 0.003763506 | 0.004751242 |
| Friend | 61 | 74 | 0.003702804 | 0.004729127 |
| School | 60 | 72 | 0.003642103 | 0.004707106 |
| Dormitory | 59 | 72 | 0.003581401 | 0.004628654 |
| Hairdryer | 56 | 63 | 0.003399296 | 0.00459043 |
| Suitable | 54 | 65 | 0.003277892 | 0.004369536 |
| Design | 53 | 64 | 0.003217191 | 0.004316287 |
| Activity | 51 | 58 | 0.003095787 | 0.004295064 |

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
