# Peer review of "How Could Consumers’ Online Review Help Improve Product Design Strategy?"

_information, doi:10.3390/info14080434_

Round 1

Reviewer 1 Report

 Thank you for giving me this opportunity to review the manuscript entitled, “Research on product improvement design strategies based on user-generated content-taking online customer reviews of cordless hair dryers as an example.”

1. Abstract

The abstract needs to present the purpose, method, data, results, and implications in brief. The data and the implications are not well presented. The abstract needs to be revised.

2. Introduction

In the last paragraph of the introduction, the sequential methods are presented but the purpose of this study, objectives, and expected contributions are clearly presented and justified. The importance of this study should be highlighted.

3. Literature review

All information about literature review is about LDA. The most important content is products, hair dryers and consumers’ reviews. The literature review also did not present research questions, hypotheses. Moreover, the introduction present several methods for analyzing consumers’ reviews. The literature review seems to focus on LDA, the topic modeling approach.

4. Figure 2.

Are the authors going to use TF-IDF or any word pre-processing steps before the LDA? OR What are the other analyses such as sentimental analysis and content analysis mentioned in introduction?

5. Coding

On the coding lines are developed by the authors?

6. Table 2

What is about table 2. Cordless hairdryer design index system? The table seems not to be obtained by the LDA approach. The specific information and steps should be noted how to analyze 1405 comments.

7. Conclusions and limitations.

This study should add theoretical and practical implications. The information is very weak. And The important implications and discussions should be added. 

Reviewer 2 Report

The goal of the paper is to discuss the Product Improvement Design Strategies Based on User-Generated Content-Taking Online Customer Reviews. The paper is overall well written, but it needs some improvement.

1. The title of the paper is too specific. There is no need to mention Cordless Hair Dryers as an Example in the title.

2. The abstract is written too technically, and in a generic manner. Algorithms and software are too extensively mentioned in the abstract, they are just technical tools and are not highly relevant for the research. There is no information about the Product Improvement Design Strategies in the abstract. Theorethical and practical contributions of the paper are not mentioned in the abstract, and they should be.

3. Introduction is well written, with sufficient and relevant information. However, there should be a  paragraph in which the theorethical contributions and practical implications are elaborated, with references. Suggested references are: Nguyen, B., Nguyen, V. H., & Ho, T. (2021). Sentiment analysis of customer feedbacks in online food ordering services. Business Systems Research: International journal of the Society for Advancing Innovation and Research in Economy12(2), 46-59.

Grosu, V., Brinzaru, S. M., Ciubotariu, M. S., Kicsi, R., Hlaciuc, E., & Socoliuc, M. (2022). Mapping Future Trends in Integrated Reporting, CSR and Business Sustainability Research: A Cluster-based Approach. ENTRENOVA-ENTerprise REsearch InNOVAtion8(1).

Chun, H., Leem, B. H., & Suh, H. (2021). Using text analytics to measure an effect of topics and sentiments on social-media engagement: Focusing on Facebook fan page of Toyota. International Journal of Engineering Business Management13, 18479790211016268.

4. The code examples should be moved to Appendix. 

5. Is it legal to crawl the data from this specific website? Did you obtained permission of the website owner. This should be mentioned in the paper. 

6. I suggest that you divide the Methodology section into four steps, and describe each step in a sub-chapter. The text would be easier to read. Also, provide a better description of each of the steps. Mathematical equations for the algorithms used should be also mentioned. At the end, please, mention the limitations of the used approach.

7. Remove Figure 1 from the paper. It gives unprofessional impression. Instead, you can present the print screen of the webiste that was scrapped. Also, if you decide to present it, obtain the permission from the website owner. 

8. Is it appropriate to use the term "Naming"? I think that "Labeling" would be more appropriate. 

9. Figure 6 is not useful in the paper, since it presents the word cloud in chinese. The reader would not be able to understand it. Either prepare thew word cloud in English or remove it. 

10. Figure 2 is of low quality. 

11. What was the theorethical framework for the development of the Table 2. In other words, how did you come up with the categories in the table? You should explain this in the Methodology.

12. In the last section, please focus on “Discussion, Implication, and Conclusion” to include

(1).     Summary of the research - what was the goal, and how was it attained (2).     Theoretical implications - Discuss why the authors found these results and how they comply (or do not) with the Literature Review. (3).     Managerial Implications - Discuss why and how your results are relevant to the practice. (4).     Limitations of the paper (5).     Future Studies and Recommendations    

Overall, english language should be corrected. Strange terms are used, e.g. naming instead of labeling, Establishment of Cordless Hairdryer Design Index (it should be development), and so on. 

Reviewer 3 Report

The authors deal with an important issue that is worthy of examining. However, they should detail on how the customer reviews of cordless hairdryers are done on JD.com and how the data are collected and screened for further analyses. Also, a reporting and discussion on the related statistics of the data will help increase the credibility of the research. In addition, the exposition of the paper is below the publication standard and needs improvements by the authors. For instance, we often avoid using the wording “research on” and “as an example” in the title of an academic paper.

The exposition of the paper needs significant improvements by the authors. 

Round 2

Reviewer 1 Report

Thank you for revising the manuscript. Overall, I think the contribution is limited in academia. The method is somewhat weak, and the literature review did not present a guiding theory or propose clear research questions based on previous research. The outcomes are not well presented because the two methodological approaches are not fully conducted.

1. Since the coding is not developed by the authors, the coding in Appendix is not necessary. I also cannot see the clear references. The Appendix A should not be presented in the current form because the coding is not authors’ original work.

2. The reviews are written in Chinese. Therefore, a single word of Chinese is translated into two words in English as shown in Appendix B. I would like to suggest adding Chinese next to the translated words. Presenting both original and translated words would be helpful for audiences.

3. Method

I found two approaches such as LDA and in-depth interview. Two approaches are not fully met the rigorous standards of academic research, but the contents and the procedures appear to be a small project for writing a report for hair dryer companies.

Author Response

Dear My Best Reviewer,

Thank you very much for your valuable advice.

I tried to follow your instructions because I was so impressed by the details you provided (we were really impressed). I recognize that this article is far from perfect, but hopefully, it is now more systematic and easier to read. I am very grateful for your interest in my article. I hope to become a good reviewer like you one day. Thank you for your valuable advice.

Best Regards,

Note: Please see the attachment

Reviewer 3 Report

The authors have mitigated the concerns I raised in the previous's rounr review. That said, I feel the writing and exposition of the paper can be improved further. I leave this for the authors to do with and make the paper to the best possible before publication. 

The writing and exposition of the paper can be improved further. 

Author Response

Dear Reviewer,

Thank you very much for reviewing our paper again and acknowledging the modifications we made. We are pleased to know that you recognize our careful attention to the comments you provided in the previous round and the efforts we made in addressing them.

We have made diligent revisions to the paper's writing and exposition, aiming to make it more clear and coherent. We considered your suggestions and made corresponding adjustments in the manuscript to ensure that readers can better understand our research findings.

Once again, we appreciate your guidance and support. We value your professional insights as they are crucial to improving the quality of our research and refining the paper. We believe that your valuable advice will help elevate the paper to a higher standard, providing readers with more valuable content.

If there are any other aspects that need improvement or further adjustments, please do not hesitate to let us know. We look forward to working with you again to further enhance the paper before publication.

Thank you once more for your sincere efforts,

Wei Miao